# Assessing the Use of BotulinumtoxinA for Hyperactive Urinary Tract Dysfunction a Decade After Approval: General Versus Local Anesthesia for BotulinumtoxinA Detrusor Injection

**DOI:** 10.3390/toxins16110462

**Published:** 2024-10-28

**Authors:** Heinrich Schulte-Baukloh, Apostolos Apostolidis, Catarina Weiss, Thorsten Schlomm, Sarah Weinberger, Dirk Höppner, Kathrin Haberecht, Carsten Waskow, Hendrik Borgmann, Jörg Neymeyer, Bernhard Ralla

**Affiliations:** 1Department of Urology, Charité—University Hospital Berlin, 12203 Berlin, Germany; 2Urologic Practice Turmstrasse, Mitte/Moabit, 10551 Berlin, Germany; 3Department of Urology, University Hospital Brandenburg, 14770 Brandenburg, Germany; 42nd Department of Urology, Aristotle University of Thessaloniki, 56403 Thessaloniki, Greece; 5Urologic Practice Kurfürstendamm, Charlottenburg, 10711 Berlin, Germany

**Keywords:** overactive bladder, neurogenic bladder, botulinumtoxinA detrusor injection, pain, anxiety, local anesthesia, general anesthesia

## Abstract

**Background**: The onabotulinumtoxinA detrusor injection (OnabotA DI) was approved a decade ago for the treatment of patients with idiopathic overactive bladder (iOAB) or neurogenic detrusor overactivity (nDO) dysfunction who had not been treated successfully otherwise. The procedure is usually performed under local anesthesia (LA), and various approaches have been investigated to make the procedure as painless as possible. We examined the level of anxiety and pain experienced by patients who wanted to have the procedure performed under LA or general anesthesia (GA). **Material and Methods**: Patients scheduled for OnabotA DI were able to choose the anesthesia procedure (LA or GA). The Amsterdam Preoperative Anxiety and Information Scale (APAIS) was used to grade anxiety before anesthesia or before the procedure itself. Intra- and postoperative pain was determined using the Visual Analogue Scale (VAS). Various established questionnaires (including the Urinary Distress Inventory UDI-6), as well as a postoperative satisfaction questionnaire, were used to evaluate the success of the therapy. **Results**: In total, 104 patients (93 F, 11 M; age 64.0 (22–89) years; 80× iOAB, 24× nDO) were evaluated. OnabotA-DI was performed with LA in 72 patients and GA in 32. Stratified by first versus repeat injection in the LA group, there was a significant decrease in the Anxiety Score in the first vs. repeat injection group (*p* = 0.038). The LA group showed higher concerns in the anesthesia questions of the Amsterdam Preoperative Anxiety and Information Scale (APAIS) than the GA group (OR: 0.29, 95%CI: 0.02–1.74). The VAS Pain Score during the procedure was significantly lower in the GA group compared to the LA group (LA: 3.3 ± 2.2, GA group 1.5 ± 1.5; *p* < 0.001). There were no differences in the success of therapy. Despite the fear and pain, patients preferred LA to GA. **Conclusions**: This study shows that the anxiety and pain burden of patients undergoing OnabotA-DI under LA is significant in comparison to GA during the first injection, but insignificant for following injections. Overall, LA is favored over GA.

## 1. Introduction

OnabotulinumtoxinA detrusor injection (OnabotA-DI) is used for various disorders associated with an hyperactive lower urinary tract, such as idiopathic overactive bladder (iOAB) and neurogenic detrusor overactivity (nDO) in multiple sclerosis or spinal cord patients, as well as nDO of other origins, such as post-stroke or Parkinson’s disease [1], and children with spina bifida [2], or lower urinary tract symptoms due to benign prostatic hyperplasia [3], interstitial cystitis [4], and dysfunctional voiding [5]. The Food and Drug Administration (FDA) approved OnabotA-DI for nDO in multiple sclerosis (MS) patients or subcervical spinal cord patients adults or children >5 years of age and as a second-line treatment of iOAB with symptoms of urge urinary incontinence, urgency, and frequency in adults who have an inadequate response to or are intolerant of an anticholinergic medication [6].

Briefly, Botulinum-neurotoxinA (BoNT/A) cleaves SNAP-25 (synaptosomal-associated protein with a molecular weight of 25 kDa), which is, together with synaptobrevin (a vesicle-associated membrane protein) and syntaxin, an essential component of the SNARE (soluble N-ethylmaleimide-sensitive fusion attachment protein receptor) complex of the neuronal end plate. This leads to blockage of the fusion of the transmitter-containing vesicles with the presynaptic membrane. By inhibiting the release of various transmitters (e.g., acetylcholine, adenosine-triphosphate (ATP), calcitonin gene-related peptide (CGRP), or glutamate), this has an inhibitory effect on the signal transduction of efferent and afferent neuronal pathways. When injected in the bladder, both the motor cholinergic—via detrusor muscarinic receptors—and the sensory purinergic (via P2X3 receptors) signaling pathways are hindered, further to a decrease in the suburothelial afferent receptor transient receptor potential vanilloid 1 (TRPV1). An indirect way of reducing the latter receptors through inhibition of the bladder nerve growth factor is suspected. This afferent desensitization effect explains the good impact not only in patients with detrusor overactivity but also in patients with increased bladder sensation or bladder pain in interstitial cystitis/bladder pain syndrome [7]. Retrograde transfer of the toxin to the central nervous system after bladder injection also seems plausible, and thus a central effect on bladder control has been proposed as an additional mechanism of action [8].

After an initial dose-finding study [9], relief of hyperactive bladder dysfunction by OnabotA-DI for iOAB and nDO was shown in several studies [10,11,12,13,14,15]. In the approval studies [10], as well as in follow-up studies [11], regarding OnabotA-DI for iOAB, there was a reduction in daily incontinence episodes in the OnabotA group compared to the placebo group, a corresponding reduction in the mean number of daily micturition, urgency, and nocturia episodes, and a significant, positive influence on the patients’ quality of life [15]. Recent data (GRACE-study [16]) also confirm these results in the real world, as well as impacting the use of incontinence aids and medication [17]; there was also a significantly lower incidence of urinary retention requiring catheterization, in contrast to 5.8% in the approval study [10]. Patients with nDO showed a significant reduction in urinary incontinence, an increase in bladder capacity, an increase in urodynamic reflex volume, and a reduction in detrusor overactivity after 200 or 300 U of OnabotA DI [13]. Side effects in both nDO and iOAB were summarized in a recent meta-analysis [18].

The procedure of OnabotA-DI is carried out cystoscopically, usually under local anesthesia (LA), by injecting OnabotA dissolved in sodium chloride into approximately 20 locations of the detrusor muscle, including the side walls, the posterior wall, and the base of the bladder. Most surgeons include the trigonum, contrary to the manufacturer’s instructions. A needle with a length of 4–5 mm and a diameter usually of 22–27 gauge is used for the injection.

However, for many patients, a major hurdle to choosing this therapy to relieve their symptoms is fear of the invasive procedure, cystoscopy, multiple injections, and side effects such as urinary retention. In particular, pain is an obstacle for patients who may undergo this therapy for the first or repeated times, and surgeons alike might have concerns about inadequate local anesthesia [19]. Accordingly, there are a number of approaches to positively influence the pain of injection therapy. For example, fewer injections are carried out than the 20 injections specified by the manufacturer, apparently without any loss of effectiveness [20] (but there are also contradictory results in terms of pain diminution by reducing the injection number [21]). Sodium bicarbonate may be added to the anesthesiological lidocaine solution because increasing the pH value increases the permeability of the urinary bladder mucosa for the LA, thus contributing to a reduction in pain [22]. In rare cases, electromotive drug administration has also been used successfully [23].

Due to the fear of pain during the procedure, there are patients who prefer the procedure to be carried out under general anesthesia (GA). In order to obtain a clearer picture of the stress on patients under LA versus GA, we examined whether patients generally prefer to undergo OnabotA DI under LA or GA, what the different levels of anxiety are before anesthesia or before the surgical procedure, what the different levels of pain are, whether there are differences in the treatment outcome, and which form of anesthesia is preferred by patients who experienced both forms of anesthesia.

## 2. Results

In total, 104 patients (93 F, 11 M; mean age: 64.0 (22–89) years, median 68 years) were evaluated (Figure 1); 80 patients had iOAB and 24 had nDO. OnabotA DI was performed in 72 patients with LA and in 32 with GA. The groups were comparable with regard to age and type of bladder dysfunction. Male patients decided significantly more often to have the procedure performed under GA (Table 1).

The following results were found regarding undergoing OnabotA DI under LA versus GA. The preoperative APAIS showed no significant differences in the sum score between the two groups in the category of anxiety in the overall cohort (initial and repeat injections together; total score LA group 6.5 ± 3.5 vs. GA group 6.0 ± 3.6; *p* = 0.4). However, individual questions in the APAIS questionnaire showed that the question about fear of anesthesia in the LA group was given a score ≥ 4 (very great or extreme fear) by one in ten patients (10.3%), in contrast to 3.2% in the GA group (odds ratio (OR): 0.29, 95% CI: 0.02–1.74). The situation was similar (score ≥ 4) for “being concerned about the anesthesia” (LA vs. GA: 7.8% vs. 3.2%; OR: 0.39, 95% CI: 0.02–2.58). The “Desire for Information Score”, a query in the APAIS questionnaire that assesses the need for the extent of explanation/information, was not significantly different (Table 2). Similarly, fear of the OnabotA-DI procedure itself was not different between groups.

However, when stratified according to whether the procedure was performed for the first time (LA group: 30; GA group: 10) or the repeated time (LA group: 40; GA group: 21; unknown: 3), the Anxiety Score in the LA group at the first injection was significantly higher than in the GA group (8.7 ± 4.2 vs. 5.9 ± 3.0; *p* = 0.038; Table 3, visualized in Figure 2). In the LA group, the Anxiety Score for the repeat injection was highly significantly lower than at the first injection (8.7 ± 4.2 vs. 4.8 ± 1.7; *p* < 0.001). The Anxiety Score for the repeat injection did not differ significantly between the LA and GA groups (*p* = 0.2; Table 3).

In the GA group, fear scores were not significantly different between the first and repeated injections (5.9 ± 3.0 vs. 6.2 ± 4.0, *p* = 0.816). However, there was a significant difference in the Pain Score during the procedure (Figure 3). Patients in the GA group tended to report that the procedure was more unpleasant within the first 24 h after the procedure than those in the LA group (GA group: 2.9 ± 2.4 vs. LA group: 2.2 ± 1.7, n.s.). Information about the most severe postoperative pain or the use of painkillers was not significantly different between groups.

The incontinence questionnaires 4 weeks after the procedure were completed by 55% of the patients (*n* = 57; therapy outcome was no primary variable). All changes in the questionnaires showed highly significant (*p* < 0.001) improvements in symptoms but without significant differences between the LA and GA groups. The changes in this cohort showed a decrease in the Urinary Distress Inventory-6 (UDI-6) score (0–100) from 56.6 to 22.8, in the Incontinence Impact Questionnaire-7 (IIQ-7) score (0–100) from 64.9 to 23.4, in the Symptom Severity Index (SSI) score (0–20) from 11.7 to 5.5, and in the Symptom Impact Index (SII) score (0–12) from 5.1 to 1.5. “Complications from the injection” were not reported by patients in the LA or GA groups. Overall satisfaction with the injection (score 0–10) showed scores of 7.3 ± 3.2 (LA) vs. 8.2 ± 2.6 (GA) (*p* = 0.24). An all-over better bladder/urine control was reported by 86.5% in the LA group and 94.7% in the GA group (*p* = 0.77). Of patients who underwent OnabotA-DI, 84.2% (LA) vs. 94.7% (GA) stated that they would choose this therapy for bladder dysfunction again; the difference was not significant (*p* = 0.54).

Of patients who stated that they had had the procedure performed with both anesthesia methods in the past (*n* = 56), 37 (66%), patients stated that they preferred LA to GA, whereas 19 (34%) preferred GA to LA.

## 3. Discussion

The OnabotA-DI procedure is well-established in the treatment of hyperactive urinary bladder dysfunction of neurogenic and non-neurogenic origin. The GRACE study was able to report real-world data that documented high patient satisfaction [17].

These results were confirmed in our study using incontinence and quality of life questionnaires 4 weeks after OnabotA-DI. All incontinence scores reflected highly significant improvements. The results did not differ depending on whether the procedure was performed under LA or GA. The results of a satisfaction questionnaire, including questions about complications and choosing this therapy again, also did not differ between groups. As a result, we propose that centers delivering OnabotA-DI can give patients more freedom to choose the type of anesthesia with regard to the expected therapeutic success.

A major obstacle for patients to trust OnabotA-DI is the fear of the procedure, cystoscopy, and especially pain from the injection.

In our study, patients were able to decide for themselves which type of anesthesia they would prefer, LA or GA. Of course, this presents a bias, as GA was primarily chosen by patients who are either very sensitive to pain or have a particular fear of potential pain. To measure this fear before anesthesia or before the procedure, we used the APAIS score. The analysis of the reliability of the APAIS score revealed that the scales were reliable despite their brevity [24] and that the APAIS is also reliable and valid in the German language [25].

In patients who underwent both anesthesia procedures, approximately two-thirds preferred LA and one-third GA. Contrary to popular belief that this procedure is almost always carried out under LA, the frequency with which it is carried out under LA or GA depends on which procedures the patient can choose from. Not all institutions, especially outpatient practices, have an anesthesiologist on staff. In Germany, the procedure under inpatient conditions (where anesthesiologists will always be available) is not reimbursed (with exceptions), so that in the outpatient setting, one has to rely almost exclusively on LA. However, the fear of pain and the different levels of perceived pain should, if possible, be addressed individually by an operating doctor.

An important result of this study is that the Anxiety Score depends on whether the procedure is being carried out for the first time or for a second (or more) time. This finding may contribute to persuading the patient in the primary information discussion to have the procedure carried out under LA, especially since the effort involved in GA is, both for the patient (e.g., fasting before the GA and being accompanied home after the procedure) as well as for the department carrying out the work (e.g., providing an anesthesiologist, operating theater capacity, and costs), much higher.

It is important that the goal must be to allay the fear of significant pain in LA patients, especially before the first injection. The average pain in the LA group in this patient cohort was 3.3 ± 2.2 on the Visual Analogue Scale (VAS), which corresponds to mild to moderate pain. Further effort should be made to make the procedure more comfortable for patients. One possibility would be to allay the patient’s fear of “the uncertain procedure”. There are successful measures for this in urology. Karalar et al. [26] recently reported a significantly lower feeling of fear and pain in patients during ureteroscopy after presenting the patient with a video of the procedure instead of just verbal communication and information. The group of patients who were presented with an educational video exhibited lower anxiety levels than the patients who only received an educational interview, as reflected by the APAIS scores for anesthesia (*p* = 0.02), surgery (*p* < 0.001), overall (*p* < 0.001), and information needs (*p* < 0.001). An educational video of the OnabotA-DI would be an easy tool to set up in outpatient practice.

Pain-relieving measures also include reducing the number of injection sites while maintaining the same effect, alkalinizing the lidocaine solution [27], reshaping the injection needle [28], and, if necessary, administering further analgesia that has already been applied preoperatively.

## 4. Conclusions

This study shows significantly different levels of anxiety and pain in patients undergoing OnabotA-DI under LA or GA. Patients’ concerns about the perhaps insufficient effect of LA, especially when it is carried out for the first time, should be posed and reflected on by the doctor. In patients who underwent repeated OnabotA-DI, there were no differences in preoperative anxiety levels between LA and GA. However, the different pain levels between LA and GA remained at repeated injections. Since patients prefer LA despite the higher stress level in comparison to GA, future attention must be paid to reducing anxiety before the initial treatment (e.g., through demonstration videos) and to reducing pain through a variety of approaches.

## 5. Materials and Methods

**Patients:** Patients who were scheduled for an OnabotA-DI decided for themselves which anesthesia procedure should be used, resulting in two groups: the local anesthesia group (LA) and the general anesthesia group (GA). The indication was for iOAB or nDO, usually due to multiple sclerosis and an expanded disability symptom score of <6, but always with preserved bladder sensitivity and spontaneous bladder emptying (no self-catheterizing patients). In accordance with the requirements of the European guidelines [29], a specific history was carried out including a urine test using urine sediment and a culture to rule out a urinary tract infection or hematuria; an ultrasound with a full bladder to rule out stone disease or a tumor; and a residual urine measurement post micturition. In addition, a vaginal examination was carried out in women to rule out a localized anatomical pathology, such as prolapse, cystocele, or stress incontinence. A digital rectal examination was carried out in men and the Prostate Specific Antigen (PSA) value was measured to rule out prostate cancer.

**Measures I:** To evaluate the level of anxiety and pain, the APAIS score (4–20; [24,25]) was given during the consultation and the completed form was brought on the day of the operation. To calculate the fear and information score of the APAIS, only completed questionnaires were evaluated.

**Procedure:** Anticoagulant medication was discontinued in a timely manner according to the information in the package insert; acetylsalicylic acid was continued if necessary. On the day of the injection, the patients remained fasting under GA following the anesthesiologist’s instructions. Antibiotic prophylaxis was given preoperatively, usually trimetoprim, which was continued on the evening of the day of the operation and the next morning. Regardless of the anesthesia procedure, the operation was carried out in an outpatient setting. The injection was carried out in the lithotomy position after disinfection and sterile draping. A 21 Char. cystoscope was used with the help of an Albaran (Storz Endoskope, Knittlingen, Germany). According to the study protocol, the injection needle used was a standardized 22-gauge 4 mm needle with a 35 cm long cannula, matching the length of the cystoscope. BotulinumtoxinA was administered in doses of 100, 150, or 200 U of the preparation OnabotulinumtoxinA (Botox^®^), which was dissolved in 10 mL NaCl. For patients in the GA group, the procedure was carried out immediately after the general anesthetic. In the LA group, the urinary bladder was emptied via a single-use catheter; then, 50 mL lidocaine 2% was instilled, and this lasted for 20 min. The injection scheme included the side walls, posterior wall, bladder base, and trigone. Any minor bleeding that may have occurred was resolved before the procedure was completed. The patients remained in the recovery room of the outpatient clinic for 30–60 min before they were able to leave the practice in the presence of an accompanying person.

**Measures II:** To record pain, the VAS pain score from 1 to 10 was marked. This VAS score was asked twice in writing to represent intraoperative pain and pain within 24 h after the procedure. The success of the therapy was measured by incontinence questionnaires administered preoperatively and 4 weeks postoperatively (Urinary Distress Inventory UDI-6 (score 0–100): Incontinence Impact Questionnaire IIQ-7 (score 0–100), Symptom Severity Index SSI (score 0–20), and Symptom Impact Index SII (score 0–12)), as well as a general postoperative satisfaction questionnaire, which was aimed, among other things, at pain treatment and the use of painkillers.

The study protocol was approved by the ethics committee of the Universitätsmedizin Berlin (ethics vote number EA4/203/22, ethics committee of the Universitätsmedizin Charité Berlin, Benjamin Franklin Campus).

## 6. Statistical Tests

Continuous variables such as age were analyzed by calculating the mean, median, standard deviation, and range. Tests for differences between groups were based on either *t*-tests (normally distributed variables) or Wilcoxon’s rank-sum test. Categorical variables were characterized by calculating absolute and relative frequencies. Corresponding statistical tests used were Fisher’s exact test or the Chi-square test. All analyses were performed with R Core Team (2023). _R: A Language and Environment for Statistical Computing_. R Foundation for Statistical Computing, Vienna, Austria. “R version 4.3.1 (16 June 2023 ucrt)”, RStudio Team (2020). RStudio: Integrated Development for R. RStudio, PBC, Boston, MA, USA.

## Figures and Tables

**Figure 1 toxins-16-00462-f001:**
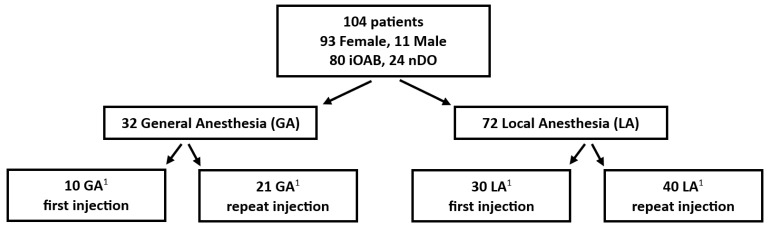
Overview of the study protocol (^1^ difference to 104: *n* = 3 unknown whether first or repeated injection).

**Figure 2 toxins-16-00462-f002:**
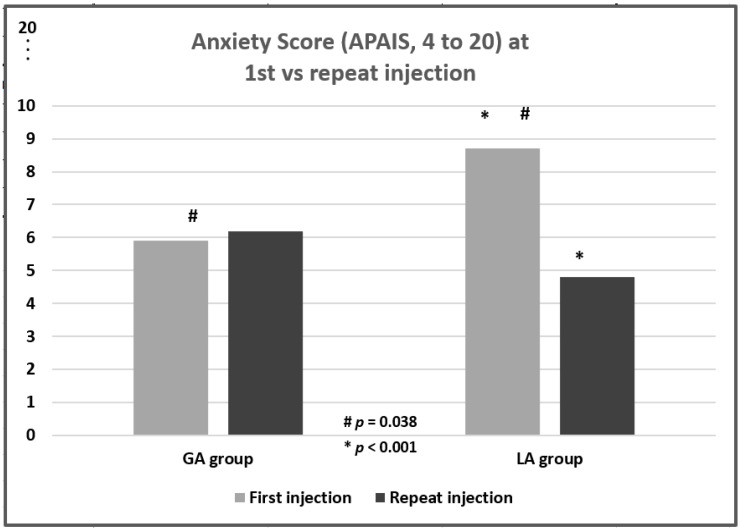
Anxiety Score of the APAIS Questionnaire (Amsterdam Preoperative Anxiety and Information Scale), stratified for first versus repeat injection in the LA and GA groups.

**Figure 3 toxins-16-00462-f003:**
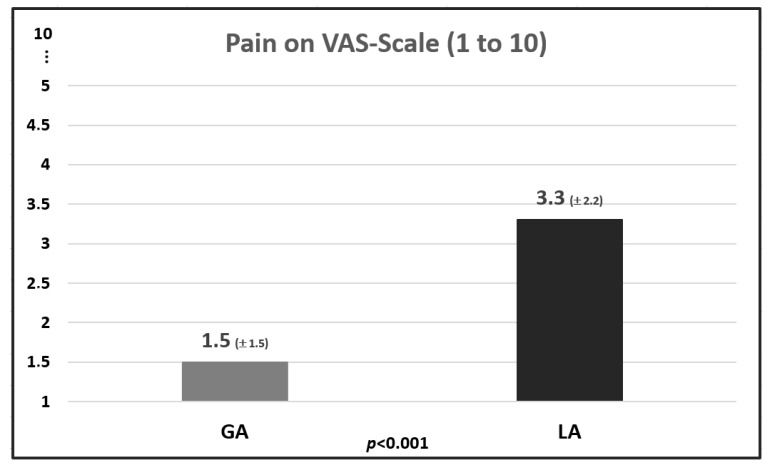
Pain Score on Visual Analog Scale (VAS, can be marked from 1 (no pain) to 10 (worst pain imaginable)) during OnabotA detrusor injection in general anesthesia (GA) and local anesthesia (LA) patients. Values are given in mean (±standard error).

**Table 1 toxins-16-00462-t001:** Basic demographics, diagnosis, and dosage of OnabotulinumtoxinA (OnabotA); GA = general anesthesia, LA = local anesthesia; standard deviation (SD); male (M), female (F); idiopathic overactive bladder (iOAB), neurogenic detrusor overactivity (nDO); ^1^
*n*; ^2^ Wilcoxon rank-sum test; Fisher’s exact test; Pearson’s Chi-squared test.

Characteristic	Overall, *n* = 104 ^1^	GA, *n* = 32 ^1^	LA, *n* = 72 ^1^	*p*-Value ^2^
**Age**				0.2
Mean (SD)	64.0 (13.9)	61.9 (12.6)	65.0 (14.4)	
Median	67.5	62.0	68.0	
Range	22.0–89.0	32.0–80.0	22.0–89.0	
**Gender**				**<0.001**
M	11 (10.6%)	10 (31.2%)	1 (1.4%)	
F	93 (89.4%)	22 (68.8%)	71 (98.6%)	
**Diagnosis**				0.8
OAB	80 (76.9%)	24 (75.0%)	56 (77.8%)	
nDO	24 (23.1%)	8 (25.0%)	16 (22.2%)	
**Dosage**				0.9
100	79 (76.7%)	25 (78.1%)	54 (76.1%)	
150	9 (8.7%)	2 (6.2%)	7 (9.9%)	
200	15 (14.6%)	5 (15.6%)	10 (14.1%)	

**Table 2 toxins-16-00462-t002:** Anxiety Score and Desire for Information Score of the APAIS Questionnaire (Amsterdam Preoperative Anxiety and Information Scale).

Characteristic	Overall, *n* = 104	GA, *n* = 32	LA, *n* = 72	*p*-Value
**Anxiety Score**				0.4
Mean (SD)	6.3 (3.6)	6.0 (3.6)	6.5 (3.5)	
Median	5.0	4.0	5.0	
Range	4.0–16.0	4.0–16.0	4.0–16.0	
Missing	18	3	15	
**Desire for** **Information Score**				0.5
Mean (SD)	4.2 (2.4)	4.5 (2.6)	4.1 (2.4)	
Median	4.0	4.0	4.0	
Range	0.0–9.0	0.0–9.0	0.0–8.0	
Missing	1	1	0	

**Table 3 toxins-16-00462-t003:** Anxiety Score and Desire for Information Score of the APAIS Questionnaire (Amsterdam Preoperative Anxiety and Information Scale) stratified for first versus repeat injection. Significant differences are shown in **bold** numbers.

	First Injection			Repeated Injection		
Characteristic	GA, *n* = 10	LA, *n* = 30	*p*-Value	GA, *n* = 21	LA, *n* = 40	*p*-Value
**Anxiety Score**			0.038 ^1^			0.2
Mean (SD)	**5.9 (3.0)**	**8.7 (4.2) ^2^**		6.2 (4.0)	4.8 (1.7) ^2^	
Median	4.0	8.0		4.0	4.0	
Range	4.0–13.0	4.0–16.0		4.0–16.0	4.0–12.0	
**Desire for** **Information Score**			0.6			0.7 ^1^
Mean (SD)	5.0 (2.6)	4.4 (2.6)		4.3 (2.7)	3.9 (2.1)	
Median	5.0	4.0		4.0	4.0	
Range	2.0–8.0	0.0–8.0		0.0–9.0	0.0–8.0	

^1^ Wilcoxon rank sum test. ^2^ Difference in LA group Anxiety Score first vs. repeat injection: *p* < 0.001.

## Data Availability

The raw data supporting the conclusions of this article will be made available by the authors on request.

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
