# Peer review of "Assessing the Use of BotulinumtoxinA for Hyperactive Urinary Tract Dysfunction a Decade After Approval: General Versus Local Anesthesia for BotulinumtoxinA Detrusor Injection"

_toxins, 2024, doi:10.3390/toxins16110462_

Round 1
Reviewer 1 Report
Comments and Suggestions for Authors
In this study, the authors examined the use of botulinum toxin injections into the bladder for patients with hyperactive bladders. The main focus was the preference in anesthesia used to perform these injections. The study's main finding is that despite higher fear and pain associated with local anesthesia (LA), it is preferred over general anesthesia (GA). This is an important finding.
It was also interesting to note that there was a significant decrease in anxiety score in the first vs repeat injections which was also a significant finding.
The statistical tests used were also described and are appropriate for this study.
Author Response
Please see the attachment. Further, you find the red-tracked version under "manuscript", and the final clean version under "supplement".

Reviewer 2 Report
Comments and Suggestions for Authors
Dear Authors,
Manuscript entitled „ Assessing the Use of BotulinumtoxinA for Hyperactive Urinary Tract Dysfunction a Decade after Approval: General versus Local Anesthesia for BotulinumtoxinA Detrusor Injection” is an interesting, well-written and well-planned experimental work. I fully support the publication of this manuscript; however I recommend some small corrections according to the comments:
Abstract
Line 15 – explain in full name an abbreviation VAS
Line 21 - explain in full name an abbreviation APAIS
Introduction
Line 39 - explain in full name an abbreviation FDA
Line 40 - explain in full name an abbreviation MS
Line 44 - explain in full name an abbreviation BoNT/A
Line 49, 50 - explain in full name abbreviations ATP, CGRP
Line 54 - explain in full name an abbreviation TRPV1
Line 67 - explain in full name an abbreviation GRACE
Results
Table 1 - explain in full name all abbreviations which appear for the first time in the description of Tables or figures: OnabotA; SD; W; M; OAB; nDO. Moreover, an abbreviation iOAB should be used instead of OAB
Line 151 - explain in full name abbreviations UDI-6, IIQ-7
Line 152 - explain in full name an abbreviation SSI score
Discussion
Line 200 - explain in full name an abbreviation VAS
Materials and Methods
Line 226 – live only an abbreviation OnabotA-DI, remove a full name
Line 237 - explain in full name an abbreviation PSA
Line 240 - live only an abbreviation APAIS, remove a full name
Line 262 - live only an abbreviation VAS, remove a full name
References
please prepare a list of publications in accordance with the rules required by the journal:
References should be described as follows, depending on the type of work:
Journal Articles:
1. Author 1, A.B.; Author 2, C.D. Title of the article. Abbreviated Journal Name Year, Volume, page range.
Author Response

(The authors gave the same response as above.)

Reviewer 3 Report
Comments and Suggestions for Authors
I have carefully review manuscript entitled “Assessing the Use of Botulinumtoxin A for Hyperactive Urinary 2 Tract Dysfunction a Decade after Approval: General versus Local Anesthesia for Botulinumtoxin A Detrusor Injection’ and found interesting, although I have some major points for betterment of manuscript:
1. Author should add schematic representation for whole study for better visualization.
2. Author should correct the references?
3. figure 2 not clearly defined, please rearrange and define correctly.
4. Please add footnote in table1 and table 2.
5.Please mention ethical approval details?
Comments on the Quality of English Language
NA
Author Response

(The authors gave the same response as above.)
